# Augmented RBMLE-UCB Approach for Adaptive Control of Linear Quadratic Systems

**Akshay Mete**
Texas A & M University
College Station, Texas, USA
akshaymete@tamu.edu

**Rahul Singh**
Indian Institute of Science
Bengaluru, Karnataka, India
rahulsingh@iisc.ac.in

**P. R. Kumar**
Texas A & M University
College Station, Texas, USA
prk@tamu.edu

## Abstract

We consider the problem of controlling an unknown stochastic linear system with quadratic costs – called the adaptive LQ control problem. We re-examine an approach called "Reward-Biased Maximum Likelihood Estimate" (RBMLE) [1] that was proposed more than forty years ago, and which predates the "Upper Confidence Bound" (UCB) method as well as the definition of "regret" for bandit problems [2]. It simply added a term favoring parameters with larger rewards to the criterion for parameter estimation. We show how the RBMLE and UCB methods can be reconciled, and thereby propose an Augmented RBMLE-UCB algorithm that combines the penalty of the RBMLE method with the constraints of the UCB method [3], uniting the two approaches to optimism in the face of uncertainty. We establish that theoretically, this method retains $\tilde{\mathcal{O}}(\sqrt{T})$ regret, the best known so far. We further compare the empirical performance of the proposed Augmented RBMLE-UCB and the standard RBMLE (without the augmentation) with UCB, Thompson Sampling, Input Perturbation, Randomized Certainty Equivalence and StabL on many real-world examples including flight control of Boeing 747 and Unmanned Aerial Vehicle. We perform extensive simulation studies showing that the Augmented RBMLE consistently outperforms UCB, Thompson Sampling and StabL by a huge margin, while it is marginally better than Input Perturbation and moderately better than Randomized Certainty Equivalence.

## 1 Introduction

Consider a linear stochastic system $x_{t+1} = A^\star x_t + B^\star u_t + w_{t+1}$, where $x_t \in \mathbb{R}^n$, $u_t \in \mathbb{R}^m$ and $w_t \in \mathbb{R}^n$ are the state, control applied, and state noise, respectively, at time $t$. We study the adaptive control/reinforcement learning problem [4, 5, 6, 7] where the controller's goal is to minimize the expected finite horizon quadratic cost $\mathbb{E} \sum_{t=1}^{T} (x_t^\intercal Q x_t + u_t^\intercal R u_t)$ by choosing the control input based on observing the past states and controls, without the knowledge of "true parameter" $\theta^\star := (A^\star, B^\star)^\intercal$. The deviation from what would have been optimally possible had the true parameter been known is measured by the "regret," [2], $R(T) := \sum_{t=1}^{T} (x_t^\intercal Q x_t + u_t^\intercal R u_t) - T J^\star(\theta^\star)$, where $J^\star(\theta^\star) := \limsup_{T \to \infty} \frac{1}{T} \mathbb{E} \sum_{t=1}^{T} (x_t^\intercal Q x_t + u_t^\intercal R u_t)$ is the optimal average cost achievable when $(A^\star, B^\star)$ are known.

36th Conference on Neural Information Processing Systems (NeurIPS 2022).

For this adaptive linear-quadratic (LQ) control problem, there are broadly four classical approaches: the Reward-Biased Maximum Likelihood Estimate (RBMLE) approach [1], the Diminishing Excitation (DE) approach [8], the Upper Confidence Bound (UCB) approach [2], and the Thompson Sampling (TS) [9] approach based on sampling from the posterior distribution. The RBMLE and UCB approaches are "certainty equivalent" (CE) in the sense that they make an estimate $\hat{\theta}_t$ of the unknown true parameter, and then take an action $u_t = K(\hat{\theta}_t)x_t$ that would be optimal if the estimate $\hat{\theta}_t$ were indeed the true parameter. They only differ in what parameter estimate they choose. The DE approach applies $u_t = K(\hat{\theta}_t)x_t + v_t$, where $v_t$ is an added "excitation," an independent noise, of diminishing variance and $\hat{\theta}_t$ is the maximum likelihood estimate. The Randomized Certainty Equivalence (RCE) [10] adds excitation to the parameter estimate.

## 1.1 The Contributions

1. We unite the two approaches to "optimism under uncertainty", RBMLE [1] and UCB [2], by showing that the RBMLE method is a penalty version of the constrained optimization problem of UCB for the case of linear quadratic systems [11, 12, 13, 14, 3]. Based on this we propose an Augmented RBMLE (ARBMLE) method that combines the penalty and constrained versions so that on the one hand it retains the analytical tractability of UCB and on the other hand provides the performance of RBMLE.

2. We determine how to choose the biasing factor $\alpha(t)$ for ARBMLE, and establish a finite time regret bound $\tilde{\mathcal{O}}(\sqrt{T})$, the same as the OFULQ algorithm of [3], the best order available to date.

3. We perform extensive comparative simulation studies of the performance of:

   (a) ARBMLE and the standard RBMLE.
   (b) OFULQ [3], which is the UCB-approach adapted to the LQ problem.
   (c) TS [15] which is the Thompson sampling approach adapted to the LQ problem
   (d) Input Perturbation (IP) [16] which is a recent reincarnation of DE for the adaptive LQ problem that additionally assumes apriori knowledge of a stabilizing controller.
   (e) Stabl [17], a modified OFULQ that adds "excitation" to the input for initialization.
   (f) RCE [10], which adds excitation to the Least Squares Estimate (LSE).

   The examples used for our simulation study have been used in many recent papers [18, 19, 17], namely (a) the longitudinal flight control of Boeing 747 with linearized dynamics [17],(b) Unmanned Aerial Vehicle (UAV) [20, 17] (c) unstable Laplacian dynamics [18], and (d) large transient dynamics [18]. Our simulation results show that ARBMLE outperforms OFULQ, TS and StabL by a large margin, which is primarily due to lack of stabilization experienced by OFULQ and TS [17]. RCE also exhibits a higher regret than ARBMLE. While the empirical performance of IP is marginally worse than ARBMLE. The results show that the ARBMLE has the same performance as the original RBMLE [21], and they both outperform all the above other algorithms. Notably the choices made by ARBMLE and OFULQ within the confidence interval are very far apart, with ARBMLE outperforming OFULQ by a large margin.

## 1.2 Previous Works

Prior work on RBMLE has concentrated on establishing long-term average optimality [22, 23, 1, 11, 24, 25, 12, 13, 14, 26, 27, 28]. Recently, its regret performance has been addressed for Multi-Armed Bandits (MABs) [29], linear contextual bandits [30], and MDPs [31]. Forced exploration techniques, somewhat similar in spirit to the $\epsilon_t$-greedy learning algorithm of [7, 32], are studied in [33, 34, 35] vis-a-vis ensuring that they do not suffer from the insufficient exploration. An adaptive LQG control based on the UCB approach, called OFULQ, is proposed in [3] which establishes a regret of $\tilde{\mathcal{O}}(\sqrt{T})$. A similar algorithm to address the adaptive LQG control problem is also designed in [36]. A computationally efficient algorithm called ROBUST with a regret of $\tilde{\mathcal{O}}(T^{\frac{2}{3}})$ is proposed in [18]. An alternative approach for designing learning algorithms is Thompson sampling [9], and [37, 38] have established an expected regret of $\tilde{\mathcal{O}}(\sqrt{T})$ in a Bayesian context. More recently, new DE-based algorithms, IP [16] and RCE [10], have been proposed that make an additional assumption, not

made in ARBMLE, OFULQ or TS, that one has access to a stabilizing controller for the unknown system, and establish $\tilde{O}(\sqrt{T})$ regret.

## 1.3 The RBMLE Approach

The RBMLE approach, proposed four decades ago [1], was the first approach not resorting to forced choices as in [39]. We begin by giving an informal description of it in the context of the adaptive LQ problem. Since the true parameter $\theta^\star := (A^\star, B^\star)^T$ is not known, one can make a Least-Squares Estimate (LSE) $\hat{\theta}_t = (\hat{A}_t, \hat{B}_t)^\intercal$ of them:

$$\hat{\theta}_t \in \text{ArgMin}_{\theta=(A,B)^\intercal} \sum_{s=0}^{t-1} ||x_{s+1} - Ax_s - Bu_s||^2. \tag{1}$$

Under the certainty-equivalence approach, the control input applied is $u_t = K(\hat{\theta}_t)x_t$ where $K(\theta)$ is the optimal linear feedback gain for the LQ problem when the system is described by $\theta = (A, B)^\intercal$. Suppose now that these estimates were to converge as $t \to \infty$ to $\hat{\theta}_\infty = (\hat{A}_\infty, \hat{B}_\infty)^\intercal$. Then, asymptotically, the input applied is $u_t \approx K(\hat{\theta}_\infty)x_t$. The closed-loop system therefore settles down to behaving according to

$$x_{t+1} \approx (A^\star + B^\star K(\hat{\theta}_\infty))x_t + w_{t+1}. \tag{2}$$

As it does so, one loses the ability to identify the matrices $(A^\star, B^\star)$, and asymptotically one can only identify the closed-loop gain $A^\star + B^\star K(\hat{\theta}_\infty)$. The problem is that as the parameter estimates begin to converge to $\hat{\theta}_\infty$, the control gain converges to $K(\hat{\theta}_\infty)$ and further exploration ceases, and one ends up only identifying the behavior of the system under the limiting gain $K(\hat{\theta}_\infty)$ being applied to the system. Since the limiting policy $u_t = K(\hat{\theta}_\infty)x_t$ need not be optimal for the long-term average cost for the true system $(A^\star, B^\star)$, the CE rule leads to a sub-optimal performance. Indeed, this problem goes by various names in different fields – the dual control problem [40, 41], the closed-identifiability problem [42, 43, 44], or exploration vs. exploitation dilemma [45].

This problem was resolved in [1] without resorting to forced exploration as in [39]. The key observation made there was that under CE the limiting parameter estimate $\hat{\theta}(\infty)$ has a one-sided bias. Specifically, $J^\star(\hat{\theta}_\infty) \geq J^\star(\theta^\star)$, i.e., the limiting parameter estimate has an optimal cost that is larger than the optimal cost of $\theta^\star$. To see this, denote by $J(K, \theta)$ the long-term cost of using the control gain $K$ when the parameter is $\theta$. Then the fact that the models $\hat{\theta}_\infty$ and $\theta^\star$ have identical behavior under the gain $K(\hat{\theta}_\infty)$ implies that $J(K(\hat{\theta}_\infty), \hat{\theta}_\infty) = J(K(\hat{\theta}_\infty), \hat{\theta}^\star)$. Now note that $J(K(\hat{\theta}_\infty), \hat{\theta}_\infty) = J^\star(\hat{\theta}_\infty)$ since the gain $K(\hat{\theta}(\infty))$ is optimal for $\hat{\theta}_\infty$. However the gain $K(\hat{\theta}_\infty)$ is not necessarily optimal for $\theta^\star$, and so $J(K(\hat{\theta}_\infty), \hat{\theta}^\star) \geq J^\star(\theta^\star)$. Therefore,

$$J^\star(\hat{\theta}_\infty) = J(K(\hat{\theta}_\infty), \hat{\theta}_\infty) = J(K(\hat{\theta}_\infty), \hat{\theta}^\star) \geq J^\star(\theta^\star). \tag{3}$$

Following from this observation, it was reasoned in [1] that if one could slightly bias the MLE to favor models with lower optimal costs $J^\star(\theta)$ so as to obtain $J^\star(\hat{\theta}_\infty) \leq J^\star(\theta^\star)$, then one would have equality throughout (3), to obtain $J(K(\hat{\theta}_\infty), \hat{\theta}^\star) = J^\star(\theta^\star)$, yielding the desired result that the gain $K(\hat{\theta}_\infty)$ is optimal for $\theta^\star$. So motivated, [1] proposed RBMLE[1] which in the context of the LSE suggests choosing the parameter estimate $\hat{\theta}_{RBMLE}(t) = (\hat{A}_{RBMLE}, \hat{B}_{RBMLE})^\intercal$ as:

$$\hat{\theta}_{RBMLE}(t) \in \text{ArgMin}_{\theta=(A,B)^\intercal} \left[ \alpha(t)J^\star(\theta) + \sum_{s=0}^{t-1} ||x_{s+1} - Ax_s - Bu_s||^2 \right]. \tag{4}$$

Generally $\alpha(t) = o(t)$ with the precise growth rate of $\alpha(t) \nearrow +\infty$ dependent on the context.

---

[1]It has been called the Cost-Biased MLE, as in [12], or Reward-Biased MLE (RBMLE) as in [29], depending on whether one minimizes a cost or maximizes a reward.

## 1.4 The UCB Approach

The "Upper Confidence Bound" (UCB) approach was first proposed in the context of Multi-Armed Bandit (MAB) Problems in [2]. In the context of Bernoulli bandits, it essentially consists of constructing, for each arm $i$ at each time $t$, a confidence interval $(\theta(i, 1; t), \theta(i, 2; t))$ of its payoff probability with confidence $(1 - \delta(t))$, for $\delta(t) = o(\frac{1}{t})$, and playing the arm $i$ with the highest value of $\theta(i, 1; t)$. This approach has been generalized to a variety of contexts including linear contextual bandits [46], Gaussian Processes [47], MDPs [48, 49], and LQ systems [3].

Particularized to LQ systems, the OFULQ algorithm of [3] that is based on the UCB approach suggests choosing $\hat{\theta}_{OFLUQ}(t) = (\hat{A}_{OFULQ}, \hat{B}_{OFULQ})^\intercal$ as the minimizer of:

$$\min_{\theta \in \mathcal{C}_t(\delta)} J^\star(\theta), \text{ where} \tag{5}$$

$$\mathcal{C}_t(\delta) := \left\{ \theta = (A, B)^\intercal : \sum_{s=0}^{t-1} ||x_{s+1} - Ax_s - Bu_s||^2 \le \gamma_T(\delta) \right\} \tag{6}$$

is a $(1 - \delta)$-high confidence set of parameters for an appropriate choice of $\gamma_T(\delta)$ [3].

## 2 The Augmented RBMLE-UCB Method

The UCB approach has been called "Optimism in the Face of Uncertainty" (OFU), since it chooses an optimistic arm after calculating the confidence intervals. The RBMLE version is also the same, though it does it in a different way by directly giving preference to parameters that can yield better rewards. It arrives at optimism in a very systematic way by noticing that closed-loop identification leads to the chain of inequalities (3), which could then be made into equalities by ensuring that $J^\star(\hat{\theta}_\infty) \le J^\star(\theta^\star)$.

We now show how one may reconcile the two approaches to optimism in the face of uncertainty in the context of the LQ problem, and then combine them to obtain a method that has experimentally superior empirical performance while also allowing a proof that it achieves the currently best known order of regret.

It can be seen that the RBMLE approach (4) can be considered an unconstrained penalty version of the constrained optimization problem (5)-(6), with a penalty factor $\frac{1}{\alpha(t)}$ for constraint violation. This provides a justification for employing "optimism" in the UCB approach.

This raises the question of whether we can take advantage of these synergies to fashion a superior algorithm, that is superior from the viewpoint of being able to provide a theoretical guarantee of the best known order of regret to date, as well as superior from the point of view of providing the best experimental performance of the algorithms to date. One can draw inspiration here from the Augmented Lagrangian Method of [50, 51]. Given a constrained problem as in (5)-(6), it adds a penalty for constraint violation, and also adjoins the constraint through a Lagrange multiplier, thus involving the constraint twice. In our case, we add the penalty and also retain the constraint. This leads to the Augmented RBMLE-UCB method:

$$\hat{\theta}_{\text{Aug}}(t) \in \text{ArgMin}_{\theta \in \mathcal{C}_t(\delta)} \left[ \alpha(t) J^\star(\theta) + \sum_{s=0}^{t-1} ||x_{s+1} - Ax_s - Bu_s||^2 \right]. \tag{7}$$

The advantage of retaining the constrained optimization is in enabling theoretical analysis, in that we can make use of the bounds on parameter estimates from concentration inequalities, as we shall show in the sequel. We thereby prove in Section 4 that it also has $\tilde{\mathcal{O}}(\sqrt{T})$ order of regret, which is the best known so far [3]. Moreover, simulations reported in Section 5 show that its performance is also much improved. In all cases, the simulation performance of ARBMLE is the same as RBMLE, raising the open problem of proving that the original RBMLE for the LQ problem also has the best known order of regret.

## 3 Problem Formulation

As introduced in section 1, we consider the following linear system,

$$x_{t+1} = A^\star x_t + B^\star u_t + w_{t+1} \tag{8}$$

where, $A^\star \in \mathbb{R}^{n \times n}$ and $B^\star \in \mathbb{R}^{n \times m}$ are unknown system parameters. Define $z_t^\top := (x_t^\top, u_t^\top)$. Then, the linear stochastic system can equivalently be written as

$$x_{t+1} = \theta^{\star\top} z_t + w_{t+1}. \tag{9}$$

The system incurs a cost at time $t$ given by, $x_t^\top Q x_t + u_t^\top R u_t$. We assume that $Q \in \mathbb{R}^{n \times n}$ and $R \in \mathbb{R}^{m \times m}$ are known positive semi-definite and positive definite matrices respectively. We make the following assumptions on $\theta^\star = (A^\star, B^\star)^\top$.

**Assumption 1.** *There exists a known positive constant $c$ such that $\theta^\star \in \mathcal{S} = \mathcal{S}_0 \cap \mathcal{S}_1$ where, $\mathcal{S}_0 = \{\theta = (A, B)^\top \mid (A, B) \text{ is stabilizable and } (A, Q^{1/2}) \text{ is detectable}\}^2$ [54], and $\mathcal{S}_1 = \{\theta = (A, B)^\top \mid trace(\theta^\top \theta) \le c\}$.*

**Assumption 2.** *Let $\mathcal{F}_t := \sigma((x_0, u_0), (x_1, u_1), \cdots, (x_t, u_t))$ denote the history of states and inputs until time $t$. The state noise $\{w_t\}$ is a martingale difference sequence with respect to $\{\mathcal{F}_t\}$, with $\mathbb{E}[w_{t+1} w_{t+1}^\top | \mathcal{F}_t] = I_n$, and is element-wise sub-Gaussian, i.e., for any $\gamma \in \mathbb{R}$, there exists $L > 0$ with $\mathbb{E}[exp(\gamma w_{t,j}) | \mathcal{F}_t] \le exp\left(\frac{\gamma^2 L^2}{2}\right) \forall j$ and $\forall t$.*

For a matrix $M$, we use $\|M\|$ to denote its operator norm induced from the $\ell_2$-norm.

## 3.1 Controller Design for a Known LQG System

For a discrete time linear system, $x_{t+1} = \theta^\top z_t + w_{t+1}$, where $\theta = (A, B)^\top \in \mathcal{S}$, there is an unique positive semidefinite matrix $P(\theta)$ that satisfies the Riccati equation (see [6])

$$P(\theta) = Q + A^\top P(\theta) A - A^\top P(\theta) B (B^\top P(\theta) B + R)^{-1} B^\top P(\theta) A.$$

The optimal control law which minimizes the long-term average quadratic cost is $u_t = K(\theta) x_t$, where the "gain matrix" is $K(\theta) := -(B^\top P(\theta) B + R)^{-1} B^\top P(\theta) A$. The optimal average cost, $J^\star(\theta)$ is equal to $trace(P(\theta))$. As a consequence of Assumption 1, the parameter set $\mathcal{S}$ is bounded, hence one can show that $P(\theta)$ is bounded as well:

$$D := \sup_{\theta \in \mathcal{S}} \|P(\theta)\| < \infty. \tag{10}$$

The following assumption is commonly made in online LQR learning problem in which the knowledge of a stabilizing controller is not made [3, 38].

**Assumption 3.**

$$\rho := \sup_{\theta \in \mathcal{S}} \|A + BK(\theta)\| < 1, \text{ and} \tag{11}$$

$$c_0 := \sup_{\theta \in \mathcal{S}} \|K(\theta)\| < +\infty. \tag{12}$$

Note that there are recent works which relax (11) slightly to $\sup_{\theta \in \mathcal{S}} \rho(A + BK(\theta)) < 1$, however they assume that the learner has access to a stabilizing controller [18, 55, 56].

## 3.2 Construction of Confidence Interval

The $\ell_2$-regularized squared fitting error with parameter $\lambda > 0$ is given by:

$$V_t(\theta) = \lambda \|\theta\|_2^2 + \sum_{s=0}^{t-1} \|x_{s+1} - \theta^\top z_s\|_2^2. \tag{13}$$

Let $\hat{\theta}_t$ be the $\ell_2$ regularized least-squares estimate of $\theta^\star$, i.e., $\hat{\theta}_t \in \arg\min_{\theta \in \mathcal{S}} V_t(\theta)$. Next, given the history $\mathcal{F}_t$, we construct a "high-probability confidence ball," $\mathcal{C}_t(\delta)$ around $\hat{\theta}_t$, i.e., a set of plausible system parameters that contains the true parameter $\theta^\star$ with a high probability. Let

$$\mathcal{C}_t(\delta) := \left\{ \theta : trace\left((\theta - \hat{\theta}_t)^\top Z_t (\theta - \hat{\theta}_t)\right) \le \beta_t(\delta) \right\}, \tag{14}$$

---

[2]We note that even though [3] requires each $\theta$ from $\mathcal{S}_0$ to satisfy the stronger reachability-observability assumption, the results therein hold true if this is replaced by the weaker stabilizability-detectability assumption [52, p.61]. In fact all that is required for the proofs to go through, is that for each $\theta \in \mathcal{S}_0$, the corresponding Riccati equation have a unique solution. It is well-known [53] that stabilizability and detectability, as above, is sufficient for this.

where $Z_t := \lambda I_{n+m} + \sum_{s=0}^{t} z_s z_s^{\mathsf{T}}$, and

$$\beta_t(\delta) := \left( nL \sqrt{2 \log \left( \frac{\sqrt{det(Z_t) det(\lambda I)}}{\delta} \right)} + \sqrt{\lambda} c \right)^2. \tag{15}$$

Define

$$\mathcal{E}_1(t) = \{\theta^\star \in \mathcal{C}_s(\delta/4), \ \forall \, s = 1, 2, \cdots, t\} \text{ and } \mathcal{E}_2(t) = \{\|x_s\| \le d_t, \ \forall s \le t\}, \tag{16}$$

where $d_t$ is defined in the Appendix. Let $\mathcal{E}_1 := \mathcal{E}_1(T)$ and $\mathcal{E}_2 := \mathcal{E}_2(T)$. Then, as in [3], $\mathbb{P}\left( \mathcal{E}_1 \cap \mathcal{E}_2 \right) \ge 1 - \frac{\delta}{2}$. Moreover, on the event $\mathcal{E}_2(t)$ defined in (16), the following holds,

$$\max_{1 \le s \le t} \|x_s\| \le X_t = Y_t^{n+m+1}, \ \forall t = 1, 2, \ldots, \text{where}, \tag{17}$$

$$Y_t := \max \left( e, \lambda(n+d)(e-1) \times \right.$$
$$\left. 4 \left( c_1 \log \left( \frac{1}{\delta} \right) + c_2 \log \left( \frac{t}{\delta} \right) \right) \right) \times \left( \log^2 \left( 4c_1 \log \left( \frac{1}{\delta} \right) + 4c_2 \log \left( \frac{t}{\delta} \right) \right) \right),$$

with $c_1$ and $c_2$ being problem dependent constants.

## 4 The Augmented RBMLE-UCB Algorithm

We employ a version of the Augmented RBMLE-UCB (ARBMLE) algorithm that proceeds in an episodic manner. Let $t_k$ denote the starting time of the $k$-th episode. Then, during episode $k$, it implements the control policy, $u_t = K(\theta_{t_k})x_t, \ \forall \, t \in \{t_k, t_k + 1, \ldots, t_{k+1} - 1\}$. where, $\theta_{t_k}$ is obtained by solving the following optimization problem,

$$\theta_{t_k} \in \arg \min_{\theta \in \mathcal{S} \cap \mathcal{C}_{t_k}(\delta)} \left\{ V_{t_k}(\theta) + \alpha(t_k) J^\star(\theta) \right\}, \tag{18}$$

where the bias-term, $\alpha(t) = \alpha_0 \sqrt{T}, \forall t$, for $\alpha_0 > 0$.

---

**Algorithm 1** Augmented RBMLE-UCB (ARBMLE)

---

   **Initialize:** $t = 0$, $Z_0 = \lambda I_{n+m}$
   **for** $k = 0, 1, \cdots$ **do**
    **if** $det(Z_t) > 2det(Z_{t_{k-1}})$ **then**
      solve the following optimization to obtain $\theta_{t_k}$,

$$\theta_t \in \arg \min_{\theta \in \mathcal{S} \cap \mathcal{C}_{t_k}(\delta)} \left\{ V_{t_k}(\theta) + \alpha(t_k) J^\star(\theta) \right\},$$

    **else**
      $\theta_t = \theta_{t-1}$
    **end if**
    $u_t = K(\theta_t)x_t$
    $Z_{t+1} = Z_t + z_t z_t^{\mathsf{T}}$
    $t \to t+1$
   **end for**

---

In Theorem 4.1, we show that regret for ARBMLE is upper bounded by $\tilde{\mathcal{O}}\left( \sqrt{T \log \frac{1}{\delta}} \right)$ which is same order as OFULQ [3].

**Theorem 4.1.** *For any $\delta \in (0,1)$ and $T > 0$, with a probability at least $(1 - \delta)$, the regret of the ARBMLE Algorithm is upper-bounded by $R(T) \le \tilde{\mathcal{O}}\left( \sqrt{T \log \frac{1}{\delta}} \right)$.*

*Proof.* Appendix A.1. □

| Ex. | RBMLE | ARBMLE | OFULQ | TS | IP | RCE | STABL |
|------|-------|--------|-------|-----|-----|-----|-------|
| (a) | 3233 | 3233 | $1.2 \times 10^6$ | $4.2 \times 10^{10}$ | 3251 | 3408 | $1.8 \times 10^6$ |
| (b) | 5930 | 5930 | $5.4 \times 10^{12}$ | $2.8 \times 10^{13}$ | 5955 | 6396 | $1.9 \times 10^{10}$ |
| (c) | 16144 | 16135 | $2.1 \times 10^{12}$ | $1.1 \times 10^{20}$ | 16164 | 180639 | $1, 2 \times 10^9$ |
| (d) | 540297 | 528805 | $4.9 \times 10^6$ | $8.2 \times 10^{11}$ | 540248 | $2.2 \times 10^{14}$ | $1.4 \times 10^7$ |

Table 1: Average Regret Performance at $T = 500$.

## 5    Empirical Performance

We evaluate the empirical performance of ARBMLE as well as standard (unaugmented) RBMLE. We compare these algorithms with OFULQ [3], Thompson Sampling (TS) [15], Input Perturbations (IE) [16], Randomized Certainty Equivalence (RCE) [10], and Stabl [17]. The results shown here are for the following examples of linear systems that have appeared in the recent literature on adaptive control of linear systems:

1. Unstable Laplacian dynamics [18, 17, 19].

2. Large transient dynamics [18].

3. Unmanned Aerial Vehicle (UAV) [20, 17].

4. Longitudinal Flight Control of Boeing 747 [17].

The details of these examples are provided in the Appendix.

Each simulation experiment is performed for a time horizon of 500 steps, and repeated 50 times. The reported results are the averaged values over the 100 runs. In Figure 1, we compare the $\log_{10}$ of the averaged regret of ARBMLE, OFULQ, TS and Stabl. In Figure 2, we plot the averaged regret of RBMLE, ARBMLE, IP and RCE. We summarize the results using the regret values at $T = 500$, averaged over the 50 runs, in Table 1. Results for more examples are provided in the Appendix. Details of the implementations can be found in the Appendix.

We highlight the key observations from above experiments:

- ARBMLE and RBMLE were always found to have the same empirical performance in most cases. More specifically, both these algorithms choose the same estimate $\theta_t$, and this suggests that $\frac{1}{\alpha(t)}$ needs to be greater than the Lagrange multiplier for the ball constraint $\theta \in \mathcal{C}_{t_k}(\delta)$. This also motivates future study of the regret of the (unaugmented) RBMLE.

- As can be seen in Figure 1, ARBMLE/RBMLE outperform OFULQ and TS by a huge margin. We conjecture that this is due to temporary instability when OFULQ, TS are employed [17]. ARBMLE/RBMLE also has a signficantly lower regret as compared with StabL.

- Figure 2 shows that ARBMLE/RBMLE also outperforms RCE moderately, and IP marginally.

**Remark 1.** *Our simulations for ARBMLE, OFULQ and TS are based on the confidence interval $\beta_t(\delta)$ as defined in* (15). *Instead, recent works [17, 18] use $\beta_t(\delta) := trace\left( (\theta^\star - \hat{\theta}_t)^\intercal Z_t(\theta^\star - \hat{\theta}_t) \right)$. However, one may note that $\theta^\star$ in $\beta_t(\delta)$ is not known to the learning agent, and so such a definition of $\beta_t(\delta)$ is not a viable implementation. The effect of the choice of $\beta_t(\delta)$ on the regret performance is shown in the appendix.*

## 6    Concluding Remarks

We reconcile the RBMLE and UCB approaches by showing that RBMLE is an unconstrained penalty version of the constrained optimization problem of UCB. Showing that UCB is a constrained version of RBMLE also explains why the optimism embodied in UCB-based schemes is justified. In particular, it is justified by the goal of nullifying the one-sided bias that results from the closed-loop identification of dynamics.

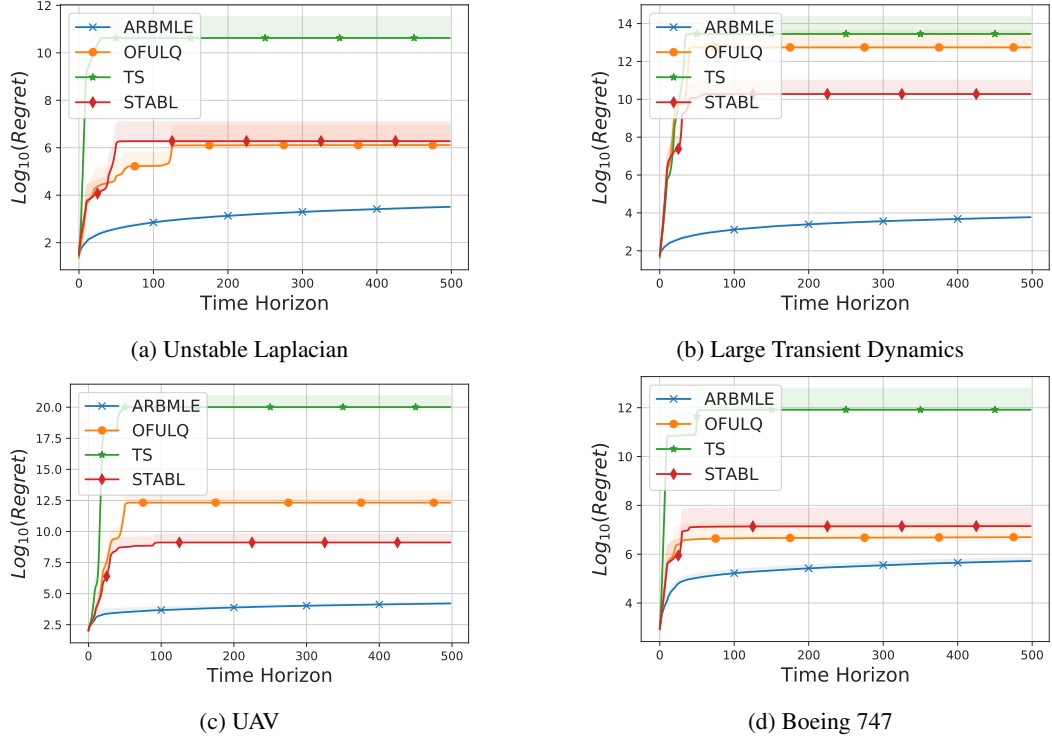

(a) Unstable Laplacian

(b) Large Transient Dynamics

(c) UAV

(d) Boeing 747

Figure 1: Logarithm of the Averaged Regret over 50 runs of RBMLE, ARBMLE, OFULQ and TS for various example systems.

Building on this, we propose an Augmented RBMLE-UCB method that not only matches the best known order of regret, $\tilde{\mathcal{O}}(\sqrt{T})$, to date – that of the OFULQ algorithm – but also outperforms OFULQ, TS and Stabl by a significant margin in simulation experiments. In fact, for the UAV experiment, the regret of OFULQ is $\approx 10^8$, TS $\approx 10^{16}$ and StabL $\approx 10^5$ times that of RBMLE. It outperforms RCE also, but the gains are moderate, while with respect to IP the gains are smaller. The simulations were carried out on real-world examples such as Boeing 747, UAV, Unstable Laplacian and Large Transient Dynamics which were taken from recent works [18, 55, 19, 17, 20] on adaptive LQG control.

This work further extends recent studies of RBMLE for MDPs [31], stochastic Multi-Armed Bandits [29] and Contextual bandits [30] establishing state-of-art regret results as well as empirically good performance.

There remain several open questions.

Currently we study only an augmented form of RBMLE, i.e. ARBMLE, in which the agent searches for a parameter value that optimizes a certain reward-biased maximum likelihood objective function within a "high-confidence ball." However, (unaugmented) RBMLE optimizes this over the larger set $\mathcal{S}$ (Assumption 1) that is known to contain the true parameter. The reason for this augmentation is simply that currently we are unable to prove regret bounds for RBMLE without it. By including it, we can capitalize on the nice technical results in [3] in the analysis of OFULQ. Simulations show that the constraint is loose, and performance is the same with/without the constraint. In fact, as shown by the simulations, the choices of $\theta_{t_k}$ made by the standard RBMLE, and the Augmented RBMLE are the same. It remains to be seen whether similar regret guarantees can be obtained by the (unaugmented) RBMLE, which remains an open problem.

One may note that ARBMLE and OFULQ differ in their choices of decisions. Denoting their estimates of the unknown parameter at a given time by $\theta_{\text{ARBMLE}}$ and $\theta_{\text{OFULQ}}$ respectively, $\theta_{\text{OFULQ}}$ always lies at the boundary of the UCB-ball constraint. This is easily seen to be true for bandits and MDPs, and empirically observed to be so for LQ systems. In contrast, $\theta_{\text{ARBMLE}}$ is most often in the interior of the UCB-ball. Moreover, while OFULQ treats all models within the UCB-ball equally and only

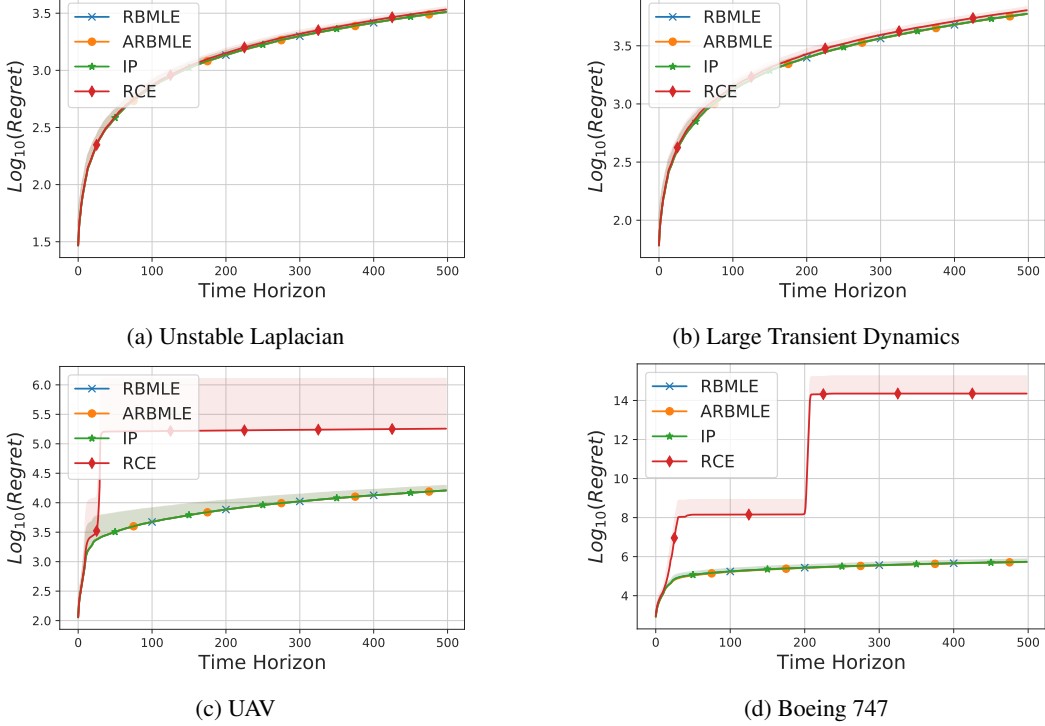

(a) Unstable Laplacian

(b) Large Transient Dynamics

(c) UAV

(d) Boeing 747

Figure 2: Logarithm of the Averaged Regret over 50 runs of RBMLE, ARBMLE, IP and RCE for various example systems. Table 1 provides the quantative results for a finer comparative evaluation of the regrets.

assesses them by their cost, ARBMLE prefers models that are closer to the Least Squares Estimate ($\theta_{\text{LSE}}$). For example, if $J^\star(\theta_1) = J^\star(\theta_2)$, then ARBMLE prefers the $\theta_i$ that is closer to $\theta_{\text{LSE}}$. We conjecture that this is the reason why ARBMLE has a significantly better performance than OFULQ – but we are unable to prove it.

It has been shown in previous works on adaptive LQG control [11, 12] that under appropriate formulations RBMLE is guaranteed to stabilize an unknown LQG system over an infinite horizon, i.e., $\limsup_{T\to\infty} \frac{1}{T} \sum_{t=1}^{T}(\|x(t)\|^2 + \|u(t)\|^2) < \infty$ a.s.. Morover, the sample path performance cost $\limsup_{T\to\infty} \frac{1}{T} \sum_{t=1}^{T}(x^\intercal(t)Qx(t) + u^\intercal(t)Ru(t))$ is (a.s.) equal to the optimal performance that could be attained if the system parameters $(A^\star, B^\star)$ were known. In fact, stability is a prerequisite before one can establish the latter type of result, as the decades of work on stochastic adaptive control in the nineteen seventies to the nineties has shown. While recent work has tended to study the finer performance measure of regret, it is usually over a finite horizon, and more attention to the stability of the learning process over an infinite horizon appears well deserved. Similarly, robustness which also was rigorously formulated and addressed in earlier decades needs to be re-examined [57].

## Acknoweldgments

This material is based upon work partially supported by the US Army Contracting Command under W911NF-22-1-0151, US Office of Naval Research under N00014-21-1-2385; 4/21-22 DARES: Army Research Office W911NF-21-20064 US National Science Foundation under CMMI-2038625, The views expressed herein and conclusions contained in this document are those of the authors and should not be interpreted as representing the views or official policies, either expressed or implied, of the U.S. Army Contracting Command, ONR, ARO, NSF, or the United States Government. The U.S. Government is authorized to reproduce and distribute reprints for Government purposes notwithstanding any copyright notation herein. The work of Rahul Singh was partially supported by the SERB Grant SRG/2021/002308, and PC 39010B.

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
