# OpenReview forum: "Augmented RBMLE-UCB Approach for Adaptive Control of Linear Quadratic Systems"
_NeurIPS.cc/2022/Conference — NeurIPS 2022 Accept_

### Official Review · Reviewer_narw · 2022-07-09

**Rating:** 8
**Confidence:** 5
**Soundness:** 4 excellent
**Presentation:** 4 excellent
**Contribution:** 4 excellent

**Summary:**

This paper develops a Reward-Biased Maximum Likelihood Estimate-Upper Confidence Bound (RBMLE-UCB) policy for controlling a stochastic linear system with quadratic costs, when its system parameters are not known to the agent. This problem is sometimes called the adaptive LQ control problem. The authors prove that this policy achieves the best-known square-root-T regret rate, and they empirically show via several numerical experiments that this approach is competitive against existing ones.

**Questions:**

I do not have any questions.

**Strengths And Weaknesses:**

The strengths of the paper are that the proposed policy is interesting, the theoretical analysis is excellent, and the numerical experiments are compelling and convincing in the quality of the proposed policy.

I do not think the paper has noticeable weaknesses.

---

> ### Author Response · Authors · 2022-08-02
> **Author Response to Reviewer narw**
>
> We thank reviewer narw for the encouraging feedback.

---

### Official Review · Reviewer_wX3F · 2022-07-12

**Rating:** 8
**Confidence:** 4
**Soundness:** 4 excellent
**Presentation:** 4 excellent
**Contribution:** 3 good

**Summary:**

The paper addresses the control of a fixed stochastic linear system of known order but unknown parameters with the objective of minimizing a finite horizon quadratic cost. The solution to this problem when the parameters are known (and under certain assumptions) is a well known result. However, when the parameters are unknown, various approaches have been suggested and explored. Many of these are mentioned in the introduction of the paper.

The paper re-examines an approach that first appeared in 1982 (ref [1]): Reward Biased Maximum Likelihood Estimate (RBMLE) but with a new twist. The new twist is to unite RBMLE with another approach known as upper confidence bound (CB) from 1985 (ref [2]). This allows the authors to show that RBMLE is a penalty version of of the UCB approach; This admits a finite time regret bound for the modified algorithm of $O(\sqrt{T})$, the best bound to date.

Through a series of simulations using established test-bed systems, the paper verifies that the new algorithm (ARMBLE) outperforms other methods, except RBMLE which achieves comparable performance.


**Questions:**

Line 49: OFU is used before being defined.

Line 141: "synergies to in fact fashion" perhaps simplify to "synergies to fashion"

Line 180: Space after ${\cal F}_t$,

Line 191: "the the".

Line 192: In equation on next line $min$ should be $\min$

Line 268: $\theta_{ARBMLE}$ should be $\theta_{\rm ARBMLE}$ and $\theta_{OFU}$ should be $\theta_{\rm OFU}.$


**Limitations:**

Nothing stands out. One suggestion is listed above. Since this is a somewhat specialized theoretical result. Perhaps its worth explaining in greater detail its value to the community.

**Strengths And Weaknesses:**

(S) Very well written, with clear explanations and a very solid and insightful analysis of a classic problem. Results in the best regret bound known to date.

(W) This is a somewhat specialized theory result. Perhaps its worth explaining in greater detail its value to the community.

---

> ### Author Response · Authors · 2022-08-02
> **Author Response to Reviewer wX3F**
>
> We thank reviewer wX3F for the encouraging feedback.
>
> **Worth explaining in greater detail its value to the community:** The LQR formulation is widely used to develop controllers for real-world control systems. RBMLE proposed in 1982 is the first learning algorithm for LQR systems that was shown to achieve long-term average optimality. While finite-time regret analysis of RBMLE in LQR setting still remains an open and challenging problem, this work takes a first step in that direction by proposing ARBMLE, an algorithm which retains the superior empirical performance of RBMLE along with regret guarantees of OFU.
>
> ARBMLE also provides an insight into the ``optimism under uncertainty'' principle. More precisely, it shows that a mild optimism is better than choosing the the most optimistic model within a confidence interval, as UCB does. Since UCB has been widely used in online learning for e.g., MDPs and Multi-armed bandits, this insight is potentially applicable to wide range of online learning problems.

---

> > ### Comment · Reviewer_wX3F · 2022-08-08
> > **Thank you for your comments on "value to the community"**
> >
> > Your comments are strictly technical but would still add value to the paper. One could also consider the question of what is  the value of these theoretical bounds in the first place. This is a question sometimes raised from the more empirical side of the RL community.

---

> > > ### Author Response · Authors · 2022-08-08
> > > **On theoritical bounds**
> > >
> > > Yes, very true; there are indeed two groups, not necessarily mutually exclusive.
> > > There are those who value empirical results, and those who value provable theoretical performance results.
> > > We have tried to address both sets of concerns.
> > > On the one hand, we have conducted empirical performance evaluation on all the systems tested in the
> > > literature to date, and have seen that the current algorithm appears to yield the best performance available so far.
> > > On the other hand, we have also provided rigorous performance bounds comparable to the best available so far.
> > > Perhaps it is not either one or the other; perhaps it is a happy situation when both simultaneously hold.

---

### Official Review · Reviewer_JHn3 · 2022-07-15

**Rating:** 5
**Confidence:** 3
**Soundness:** 4 excellent
**Presentation:** 4 excellent
**Contribution:** 2 fair

**Summary:**

This paper proposes RBMLE and ARBMLE algorithms for LQR with unknown dynamics. The authors point out that RBMLE for LQR can be viewed as a penalty version for constrained optimization in UCB for LQR. For theoretical purposes, additional constraints are added to the unconstrained optimization (4) in RBMLE, yielding ARBMLE. An O(sqrt T) regret bound is provided for ARBMLE. Then, several numerical experiments are conducted to show that RBMLE and ARBMLE yield similar performance and achieve better performance than existing methods.

**Questions:**

See above.

**Strengths And Weaknesses:**

The paper is very well written. The connection between RBMLE and UCB in the LQR problem is interesting and inspiring. The augmented version ARBMLE achieves the best regret order in terms of T. Numerical results show promising improvements over the existing methods. The appendices are also written clearly. The proof is standard and correct.

However, I have the following concerns and questions.

1. [Tractability] UCB-LQR is known to be intractable when the dimensionality is large due to the involved constrained optimization. Since ARBMLE also involves the constrained optimization, the proposed algorithm also suffers the tractability problem.

2. [Same numerical results for RBMLE and ARBMLE] The authors claim that numerically, RBMLE and ARBMLE always generate the same theta_t. I found this quite surprising. Does this result still hold for different choices of alpha0 and lambda and beta_t(sigma)?

3. [Dependence on dimensionality] Though the regret bound is optimal in terms of the dependence on T, the dependence on the dimensionality of the problem is not discussed. It has been shown that CE-type (certainty-equivalence) methods can achieve optimal regret even in terms of n and m. How does the regret of the proposed method compare with the optimal dimensional dependence in the literature?

4. [Numerical results] The authors provide four interesting numerical experiments to demonstrate the improved performance. However, all of the experiments consider a relatively small dimensionality.  How do the algorithms perform for larger dimensions, especially in comparison with CE-type methods?

=== Typos ===
1. Line 168: it should be \mathcal S instead of S.
2. Line 253: furtjer.

---

> ### Author Response · Authors · 2022-08-02
> **Author Response to Reviewer JHn3**
>
> We thank reviewer JHn3  for their detailed feedback.  The questions raised by the reviewer are addressed below:
>
> 1. **Tractability:** Yes, the tractability issue is the same for UCB-LQR and ARMBLE.
>
> 2. **Same numerical results for RBMLE and ARBMLE:** The same empirical performance of RBMLE and ARBMLE highlights that we are not compromising with the empirical performance of RBMLE algorithm by introducing an augmentation in order to establish theoretical regret guarantees. The performance of RBMLE indeed deviates from ARBMLE when the bias pre-constant $\alpha_0$ is very large.  However such cases are not of interest as our experiments consistently show that both RBMLE and ARBMLE perform better for smaller values of $\alpha_0$.
>
> 3. **Dependence on dimensionality:** The dimensionality dependence of ARBMLE regret upper bound is the same as that of OFU and TS. Recent work by Simchowitz and Foster, 2021 improves this dependency and proposes an algorithm that achieves the optimal dimensionality dependence. The fundamental difference is that their result depends on the algorithm having knowledge of a stabilizing controller. The algorithm uses this stabilizing controller as a ``fallback controller'' whenever the choice of the algorithm does not satisfy a certain stability criterion. Note that ARBMLE does not require knowledge of any such stabilizing controller, and provides the best known upper bound without the assumption of stabilizing controller.
> 4. **Numerical results for larger systems:** In order to ensure a fair comparison, our initial numerical experiments were focused on the examples used in recent literature. As pointed out by reviewer JHn3, these examples have a relatively small dimensionality. In our updated version, we have added numerical results for relatively larger systems. RBMLE and ARBMLE continue to exhibit better empirical performance similar to previous examples. The second best-performing algorithms in all our examples are the CE algorithms. The average regret performance at T=500 for these new experiments is provided in the following table.
>
> | n, m      | RBMLE |  ARBMLE     | OFU |  TS  | IP |  RCE      | STABL |
> | ----------- | ----------- |  ----------- | -----------  | ----------- | ----------- |  ----------- | ----------- |
> | 10,4 | 3440 | 3440 |  5954 |  14249  |   3463 | 3675 | 14098 |
> | 8,4 | 2703 | 2703 | 3920 | 4023  | 2720  | 2831 |10589   |
> | 6,6 | 2212 | 2212 |  2675 | 2722  | 2226 | 2298 | 13225 |

---

> > ### Author Response · Authors · 2022-08-08
> > **Author- Reviewer Discussion**
> >
> > Dear Reviewer JHn3,
> >
> > There is only one more day left for the author-reviewer discussion period -- we were wondering if you have gotten a chance to look over our responses and revisions, and if your concerns are resolved. We are happy to address any remaining questions/concerns.

---

> ### Comment · Area_Chair_9H5N · 2022-08-09
> **Thank you! Are you satisfied by the answers?**
>
> Dear reviewer,
>
> Thanks again for your detailed review! The authors have responded to the reviews. Have they answered your questions satisfactorily? If not, what are the remaining issues?
> If you have any further questions from them, please ask them now. The deadline for discussion between the reviewers and the authors is today (Tuesday, August 9th). Also as a courtesy to the authors, please acknowledge their rebuttal.
>
> Thank you,
> Area Chair

---

### Official Review · Reviewer_Qpy8 · 2022-07-19

**Rating:** 4
**Confidence:** 3
**Soundness:** 2 fair
**Presentation:** 3 good
**Contribution:** 2 fair

**Summary:**

This paper revisits two commonly used approaches in online adaptive LQR control: Reward-Biased Maximum Likelihood Estimate (RBMLE) and Upper Confidence Bound (UCB). The authors show that these two methods can be reconciled, and thereby propose an Augmented RBMLE-UCB algorithm that combines the penalty from the RBMLE method and the confidence interval of the UCB method. Theoretically, the authors establish a $O(\sqrt{T})$ regret upper bound. Empirically, the proposed algorithm outperforms several baselines in online adaptive LQR control.

**Questions:**

Please see the limitation section below.

**Limitations:**

Since 2011 (when the paper [3] is published), there are over 1000 papers studying online LQR control via learning-theoretical perspectives. In particular, the 2021 paper by Simchowitz and Foster "Naive Exploration is Optimal for Online LQR" clearly shows that the regret lower bound of online LQR control is $O(d_u^2 d_x \sqrt{T})$. Moreover, a "naive" CE control plus continual exploration (i.e., solve $K$ from OLS and $u=Kx+\mathrm{noise}$) exactly matches this lower bound! Therefore, the reviewer believes that a new paper needs to be novel and impressive enough to be published at a top conference since this field is already very crowded.

This paper provides a new algorithm for online LQR control with theoretical justification (a $O(\sqrt{T})$ regret bound) and empirical benchmarking. However, it is unclear whether the novelty is enough: Theoretically, the analysis framework is mainly from the classic UCB (OFU) method from [3]. The extra regularization term $\alpha J^*$ from RBMLE seems not fundamentally change the analysis. Empirically, the performance of the proposed augmented RBMLE-UCE approach seems to inherit from RBMLE. In Table 1, the regret numbers of RBMLE and the proposed method are exactly \emph{identical}! Given the fact that there is no regret bound for RBMLE, it seems that the UCB part is only for theoretical analysis, and does not contribute to the empirical performance at all.

Minor points:
1. In Eq. (6) and other related equations, please do not use $()^2$ as the norm of a vector. Please use $||\cdot ||^2$.
2. Assumption 3 is actually very strong. Stabilizable $(A,B)$ only guarantees the spectral radius of $A+BK$ is less than 1, rather than the 2-norm. Please comment on this.

**Strengths And Weaknesses:**

Strengths: the connection between RBMLE and UCB is interesting and the empirical performance of the proposed augmented RBMLE-UCB algorithm is sound.

Weakness: the theoretical analysis is mainly adopted from the analysis of the UCB method in [3], and the empirical performance is mainly from RBMLE. Therefore the contribution is limited.

---

> ### Author Response · Authors · 2022-08-02
> **Author Response to Reviewer Qpy8**
>
> We thank reviewer Qpy8  for a detailed feedback. The questions raised by the reviewer are addressed below:
>
>  1.  **Fundamental differences compared to Simchowitz and Foster, 2021 and other recent works:**
> Simchowitz and Foster, 2021 propose an algorithm that achieves the optimal dimensionality dependence. The fundamental difference is that their result depends on the algorithm having the knowledge of a stabilizing controller. This stabilizing controller is used as a ``fallback controller'' whenever the choice of the algorithm does not satisfy a certain stability criterion. In fact, the algorithms for online LQR can be categorized into two classes, depending on whether or not they require the knowledge of a stabilizing controller. Simchowitz and Foster, 2021, Dean et al., 2018, Cohen et al., 2019, Mania
> et al., 2019 require a stabilizing controller as an input to their algorithms. In contrast, our work along with OFU (Abbasi-Yadkori and Szepesvari, 2011), TS (Abeille and Lazaric, 2017 ), STABL (Lale et al., 2022) belongs to the class of algorithms that do not require any such knowledge of a stabilizing controller. ARBMLE provides the best known upper bound without assumption of stabilizing controller along with the apparently so far best empirical results.
>
> 2. **RBMLE and ARBMLE:** The exact same empirical performance of RBMLE and ARBMLE highlights that we are not compromising on the performance of RBMLE algorithm by introducing the augmentation, which is only done in order to establish theoretical regret guarantees. In other words, ARBMLE is able to achieve ``best of both the worlds.''
> The motivation behind choosing an estimate within the confidence interval is that with a high probability, the true parameter lies within the confidence interval. OFU selects the most optimistic estimate within the confidence interval, while ARBMLE prefers an estimate which is only mildly optimistic as compared with the least squares estimate (LSE). The superior empirical performance, while retaining the same provable theoretical guarantees provides the key insight that a mild optimism is better than choosing the most optimistic model. Since UCB (OFU) type algorithms have been extensively used in online learning including MDPs, and a variety of bandit settings, this insight is valuable for the wide-class of online learning problems along with the present LQR problem.
>
>  3.  **Regarding Assumption 3:**
> The reviewer apparently misunderstood our ``induced operator norm"  as the Frobenius norm. In Assumption 2, we say that  For a matrix $M$, we use $ | M | $ to denote its operator norm induced from the $ \ell_2 $-norm.''
> Later in equation (10) we say that
> $$ \rho:= \sup_{\theta \in \mathcal{S}} ||A+BK(\theta)|| <1.$$
> This is the same as saying that the spectral radius is less than 1.

---

> > ### Author Response · Authors · 2022-08-08
> > **Author- Reviewer Discussion**
> >
> > Dear Reviewer Qpy8 ,
> >
> > There is only one more day left for the author-reviewer discussion period -- we were wondering if you have gotten a chance to look over our responses and revisions, and if your concerns are resolved. We are happy to address any remaining questions/concerns.

---

> > > ### Comment · Reviewer_Qpy8 · 2022-08-09
> > > **Reply**
> > >
> > > Thanks for the response. The reviewer still has concerns:
> > > 1. In all the experiments (including the new ones in higher dimensional systems), the performance of RBMLE and ARBMLE are always identical, which suggests the confidence set could be possibly vacuous so the UCB approach actually does nothing in practice. The reviewer believes it is very important to understand it deeper. If RMBLE's performance is always the same as ARBMLE's, why the latter one has guarantees and the former one does not?
> > > 2. There is no significant novelty in the proof techniques. Most of them are directly following standard UCB literature.
> > > 3. It is correct that this paper does not need a stable controller to start. However, it is well known that learning a stable controller needs constant time steps. The reviewer pointed out the paper by Simchowitz and Foster to show that naive (simple) algorithms already match the theoretical lower bounds of the learning LQR problem, so a new paper in the field needs to either provide new proof techniques or theoretical insights or provide novel applications.
> > > 4. The reviewer doesn't interpret the 2-norm as the Frobenius norm. Please check the definition of stabilizable: $(A,B)$ is stabilizable if there exists $K$ such that the spectral radius of $A+BK$, namely $\rho(A+BK)$ is smaller than $1$. $\rho(A+BK)<1$ does not imply the induced 2-norm of $A+BK$, namely $||A+BK||$ smaller than 1.
> > >
> > > Therefore the reviewer will keep the evaluation for now but will see what other reviewers think about the authors' response.

---

> > > > ### Author Response · Authors · 2022-08-09
> > > > **Reply**
> > > >
> > > > (I) As we have mentioned in the paper, RBMLE yields the same performance as ARBMLE. However, we are unable to prove that RBMLE does so. This is an open question.
> > > > Yes, as the reviewer deduces, the UCB ellipsoidal constraint is loose. So it provides the insight that the confidence level needs to be increased further.
> > > >
> > > > (2)  Additionally we also need to control an additional term that represents the ``sub-optimism" of the RBMLE parameter estimate.
> > > >
> > > > (3) For a true comparison of empirical regret of a scheme (say Scheme A) that assumes knowledge of a stabilizing controller with the empirical regret of a scheme (say Scheme B) that does not assume knowledge of a stabilizing controller, one would need to add to the regret of Scheme A the empirical regret incurred during the phase to learn a stable controller. For Scheme B however, nothing needs to be added. This would further the advantage of Scheme B in comparison to Scheme A.
> > > > So the performance improvement of ARBMLE with other schemes that assume knowledge of a stabilizing controller would further improve.
> > > > However, this is nitpicking. The larger point is that there are alternative schemes, very different, and both very good. This result widens the foundation of the field by providing alternative controllers.
> > > >
> > > > (3)
> > > > Let $\lambda(A)$ := Spectral radius of $A$, and
> > > >
> > > > $||A||_{2 \to 2}$:= operator norm of $A$ induced from $\ell_2$ = Spectral norm of $A$.
> > > >
> > > > Then $\lambda(A) \leq  ||A||_{2 \to 2}$,
> > > >
> > > > So if we assume that $ sup_\theta ||A+BK(\theta)||_{2 \to 2}  = \rho < 1$, then the system is stabilizable.
> > > >
> > > >  (typo: we meant to say spectral norm rather than radius in the sentence ``This is the same as saying that the spectral radius is less than 1." in the response)

---

### Official Review · Reviewer_dwt7 · 2022-07-24

**Rating:** 4
**Confidence:** 1
**Soundness:** 2 fair
**Presentation:** 2 fair
**Contribution:** 2 fair

**Summary:**

The authors revisit an old approach to LQR, called RBMLE, and reconcile it with UCB-based methods. They show regret guarantees and good empirical performance.

**Questions:**

See above on the form of the regret bound

**Limitations:**

see main review

**Strengths And Weaknesses:**

I want to highlight that I am not familiar with the LQR setting, and I do not know part of the literature that the authors build on, in particular RBMLE approach to LQR.
I apologize with the authors and the meta-reviewer as I can only offer an educated guess; my review should be appropriately discounted.

The paper appears not to be written in a professional way. There are some inconsistencies both in the fonts and on the overall presentation style. As stated, the sequence of Lemmas is the regret analysis is not useful to understand what is going on. At a minimum, I believe that the paper writing would need to be improved for the paper to be accepted.

I have some trouble interpreting what the main result aims to convey. What \widetilde O hides here? Any dependence on the dimensionality of the system appears to be missing in the main regret bound. Shouldn’t there be any? I think the authors should try to clarify this point in the rebuttal. To me, this is the most troubling aspect of the paper.

While it appears the algorithm proposed here has better numerical performance, in practice I expect the constants in the construction of confidence intervals to play a leading role in this. I am unsure what choices were made here for this and other algorithms in order to obtain a comparison, but the point here is that the observed numerical performance may be justified by reasons that are different from the theoretical ones expressed in the paper.

Again, my apologies if I missed some important points in the paper.

---

> ### Author Response · Authors · 2022-08-02
> **Author Response to Reviewer dwt7**
>
> We thank reviewer dwt7 for their feedback.  The questions raised by the reviewer are addressed below:
> 1. **Inconsistencies in fonts and overall presentation style:** We have revised some inconsistencies that we found in the fonts and presentation.
>
> 2.  **Sequence of Lemmas:** We have added a brief paragraph at the beginning of Section 5, explaining the contents of the sequence of lemmas, so as to improve the readability of the section.
>
> 3. **Dependence of dimensionality in regret bound:** The dimensionality dependence of ARBMLE regret upper bound is the same as that of OFU. Simchowitz and Foster, 2021 employs an algorithm that achieves the optimal dimensionality dependence. However, there is a fundamental difference between their setup and our setup. Their result requires their algorithm to have access to a stabilizing controller. This stabilizing controller is used as a ``fallback controller'' in the event that the choice in their algorithm does not satisfy a certain stability criterion. In contrast, ARBMLE does not require knowledge of any such stabilizing controller, and provides the best known upper bound without assumption of the knowledge of a stabilizing controller.
>
> 4. **Constants in the construction of confidence intervals in numerical experiments:** The constants in the confidence interval are chosen exactly as in OFU. All hyper-parameters are chosen carefully to ensure fairness of comparison amongst various algorithms. The values of hyper-parameters used for all experiments are provided in the appendix.

---

> > ### Author Response · Authors · 2022-08-08
> > **Author- Reviewer Discussion**
> >
> > Dear Reviewer dwt7,
> >
> > There is only one more day left for the author-reviewer discussion period -- we were wondering if you have gotten a chance to look over our responses and revisions, and if your concerns are resolved. We are happy to address any remaining questions/concerns.

---

### Author Response · Authors · 2022-08-02
**Summary of Changes in the Manuscript**

We thank all the reviewers for their time and feedback. Questions raised by reviewers are addressed in individual comments to each review respectively. Following are the updates made in the manuscript in order to incorporate the suggestions made by the reviewers.

- More simulations have been added so as to include linear systems that have larger dimensions. These show that ARBMLE \ RBMLE maintains an edge over other algorithms, similar to what was observed for previously included examples. The average regret performance at T=500 for these new experiments is provided in the following table.

| n, m      | RBMLE |  ARBMLE     | OFU |  TS  | IP |  RCE      | STABL |
| ----------- | ----------- |  ----------- | -----------  | ----------- | ----------- |  ----------- | ----------- |
| 10,4 | 3440 | 3440 |  5954 |  14249  |   3463 | 3675 | 14098 |
| 8,4 | 2703 | 2703 | 3920 | 4023  | 2720  | 2831 |10589   |
| 6,6 | 2212 | 2212 |  2675 | 2722  | 2226 | 2298 | 13225 |

 -  As pointed out by reviewer dwt7, in order to improve readability of the section, we have now added a brief paragraph at the beginning of Section 5. This summarizes the sequence of lemmas.
-  All the grammatical errors and typos pointed out by reviewers have been corrected.

---

### Meta-Review · Area_Chair_9H5N · 2022-08-29

**Recommendation:** Accept
**Confidence:** Certain

**Metareview:**

Reward-Biased Maximum Likelihood Estimate (RBMLE) is an approach to balance exploration-exploitation that is based on biasing models with smaller cost functions. The paper considers an augmented version of it (ARBMLE) that confines the search of the model to the confidence set used by an Upper Confidence Bound (UCB)-like algorithm. The paper considers the Linear Quadratic Regulation (LQR) with unknown dynamics, provides a regret bound of ARBMLE, which is comparable to that of UCB-based approach. The paper empirically shows that both ARBMLE and RBMLE outperform many other methods, particularly UCB-based ones.

We have both strong support in favour of acceptance of this paper and some less enthusiastic negative reviews. After reading the paper, the reviews, and the discussions, I am inclined to accept the paper. The main reason is that the paper considers a relatively less-known approach to exploration-exploitation problem, provides reasonable analysis (even though the tools might be standard), and shows promising empirical results. However, my recommendation should not be considered as dismissing the concerns of reviewers. I believe many of them are valid. I merely put less weight on them in my evaluation compared to the negative reviewers.

Let me emphasize a few points brought by reviewers and my own reading of this work. I hope the authors consider them in the revision of their paper.

- The writing quality varies a lot. The first two sections are written clearly and have some nice insights and intuitions, but then the writing quality deteriorates. For example, Section 3.2 becomes confusing (we have E_t, E_1, E_2 with different meanings), and Section 5 becomes a series of lemmas without much insight. Sections 6 and 7 are of better quality again.

- The series the sequence of lemmas in Section 5 is not very insightful. The authors have added a paragraph at the beginning of that section, but I believe that is not enough. My suggestion is that authors either provide better intuition behind each of these lemma, or move them to an appendix.

- Be clear about the dependence of the regret bound on the dimension of the system.

- Assumption 3 requires more discussion.

- The issue of tractability of solving the required optimization problem should be discussed explicitly.

- Given that [31] (Mete et al., "Reward biased maximum likelihood estimation for reinforcement learning", 2021) solves an arguably more general problem (RL instead of LQR), a detailed comparison is needed. What are the differences in insight, proof techniques, etc.?

**Award:**

No

---

### Decision · Program_Chairs · 2022-09-14

Accept